# MHC Class II Presentation in Autoimmunity

**DOI:** 10.3390/cells12020314

**Published:** 2023-01-14

**Authors:** Irina A. Ishina, Maria Y. Zakharova, Inna N. Kurbatskaia, Azad E. Mamedov, Alexey A. Belogurov, Alexander G. Gabibov

**Affiliations:** 1Shemyakin-Ovchinnikov Institute of Bioorganic Chemistry RAS, 117997 Moscow, Russia; 2Department of Biological Chemistry, Evdokimov Moscow State University of Medicine and Dentistry, 127473 Moscow, Russia; 3Department of Life Sciences, Higher School of Economics, 101000 Moscow, Russia; 4Department of Chemistry, Lomonosov Moscow State University, 119991 Moscow, Russia

**Keywords:** autoimmune diseases, autoreactive T cells, human leukocyte antigen, major histocompatibility complex, negative selection, central tolerance, thymus, antigen presentation

## Abstract

Antigen presentation by major histocompatibility complex class II (MHC-II) molecules is crucial for eliciting an efficient immune response by CD4^+^ T cells and maintaining self-antigen tolerance. Some MHC-II alleles are known to be positively or negatively associated with the risk of the development of different autoimmune diseases (ADs), including those characterized by the emergence of autoreactive T cells. Apparently, the MHC-II presentation of self-antigens contributes to the autoimmune T cell response, initiated through a breakdown of central tolerance to self-antigens in the thymus. The appearance of autoreactive T cell might be the result of (i) the unusual interaction between T cell receptors (TCRs) and self-antigens presented on MHC-II; (ii) the posttranslational modifications (PTMs) of self-antigens; (iii) direct loading of the self-antigen to classical MHC-II without additional nonclassical MHC assistance; (iv) the proinflammatory environment effect on MHC-II expression and antigen presentation; and (v) molecular mimicry between foreign and self-antigens. The peculiarities of the processes involved in the MHC-II-mediated presentation may have crucial importance in the elucidation of the mechanisms of triggering and developing ADs as well as for clarification on the protective effect of MHC-II alleles that are negatively associated with ADs.

## 1. Introduction

The efficacy of the immune response, as well as the severity of the disease, is strongly associated with the genetics of individual patients. One of the most important genetic determinants specifying the predisposition to different diseases is the MHC locus. Recently, the undoubted existence of genetic gateways was demonstrated for viral infections such as SARS-CoV-2. In particular, distinct MHC-I and -II alleles increase the probability of the severe clinical course of the infection [1,2,3,4]. The existing cohorts of so-called “disease controllers” and “progressors” patients in HIV with confirmed genetic resistance to infection were evidently shown in numerous studies [5,6,7]. Summarizing, genome-encoded MHC alleles are decision-makers in terms of the dynamics and amplitude of the immune response, including autoimmune abnormalities.

A highly polymorphic MHC genomic region is linked with a broad variety of autoimmune pathologies, particularly characterized by the presence of autoreactive T cell clones [8]. One of the triggers of this autoreactive T cell response might be the presentation of self-antigens by MHC-II molecules in the periphery. One may say that autoimmune diseases may be initiated directly by MHC molecules due to their structural features independent of antigen presentation [9,10,11], but such speculations are beyond the scope of our review.

Processing of proteins captured outside the cell is the conventional method for generating antigenic peptides presented by MHC-II molecules on the cell surface. Structural polymorphism of the binding groove pockets determines a broad range of peptide ligands that can be presented by MHC-II molecules to CD4^+^ T cells. The repertoire of T cells is formed via positive and negative selection occurring in the cortex and the thymic medulla, respectively (Figure 1). Depending on T cell receptor (TCR) affinity or avidity to the peptide–MHC-II complex (pMHC), T cells undergo elimination or differentiation into the Treg or naïve CD4^+^ T cell lineage [12,13,14]. However, there are some additional factors influencing the T cell differentiation pathway during central tolerance establishment and potentially facilitating the appearance of autoreactive T cells: the topology of self-antigen presentation [15], the topology of recognition of TCR within the pMHC complex [16], the frequency of pMHC–TCR interaction [17], and costimulatory interactions [18]. Moreover, the modifications of antigens and the cytokine milieu might affect the presentation of antigenic fragments in the periphery and thus enhance the development of autoreactive T cells.

Extensive genotypic medical statistics indicate that certain MHC-II alleles are associated, either positively or negatively, with definite autoimmune diseases (ADs). The HLA-DRB1*01:01 (DR1) and 04:01 (DR4) alleles are positively associated with rheumatoid arthritis (RA) [19]. HLA-DRB1*15:01 (DR15) is the risk allele for multiple sclerosis (MS) [20] and Goodpasture syndrome [21], while HLA-DRB1*01:01 is protective against these two diseases [22,23,24]. The HLA-DQA1*05:01/DQB1*02:01 (DQ2.5) and HLA-DQA1*03:01/DQB1*03:02 (DQ8.1) alleles are positively associated with type 1 diabetes (T1D), while the HLA-DQA1*01:02/DQB1*06:02 (DQ6.2) allele is negatively associated with this disease [25,26,27]. Although a significant body of MHC-II association data has been accumulated, the correlation between such associations and autoreactive T cell response development has not yet been completely studied. Here, we attempt to establish several scenarios causing the emergence or lack of autoreactive T cells due to the pMHC engagement.

## 2. MHC-II Presentation of Low-Affinity Self-Antigens

MHC-II molecules are synthesized in the endoplasmic reticulum (ER); the peptide-binding groove is occupied by the invariant chain Ii (CD74 fragment) to prevent degradation and premature binding of self-antigens by the complex. As they migrate to the late endosomal compartments with an acidic environment, proteolytic enzymes shorten the invariant chain to a shorter class II-associated invariant chain peptide (CLIP) [28]. Protein antigens are proteolytically processed in endosomes as well (Figure 2). The peptide-binding groove of an MHC class II molecule has nine pockets that can accommodate certain amino acid residues of the peptide, typically stabilized by noncovalent bonds [29]. The P1, P4, P6, and P9 anchor residues interact with the groove of MHC-II, thus forming a binding register, while the remaining residues of the peptide are oriented in the opposite direction for TCR binding. Presumably, if there is a shift in the binding register between antigenic peptides and MHC molecules, the pMHC complex can interact with an entirely different TCR. 

The nonclassical MHC molecules HLA-DM (DM) and HLA-DO (DO) in humans and H2-DM and H2-DO in mice play an important role in exchanging CLIP for endosomal antigens [30]. By interacting with MHC-II, DM has the “editing” function: it ensures the binding of peptides with higher affinity for the MHC molecule compared with CLIP, as well as increases the loading/dissociation rate of the antigen, but it causes no effect on the equilibrium affinity [31,32]. Apparently, DM facilitates the presentation of antigen epitopes with the optimal binding register on MHC-II [33,34]. DO also performs an editing action, interacting with DM and regulating its catalytic function by preventing its binding to MHC-II [35]. Comparing MHC-II immunopeptidomes from two cell lines, DO knockout or not, it was shown that only the DR1+, DM+, and DO+ lymphoblastoid cell lines presented specific antigens on MHC-II [36]. Thus, DO contributes to the diversification of the MHC-II-presented antigenic repertoire. In experiments with DM-knockout mice, CLIP was not efficiently exchanged with other peptides, thus limiting the MHC-II antigen diversity and causing incomplete negative selection of CD4^+^ T cells in the thymus [37]. The repertoire of antigens presented by MHC-II can vary significantly depending on the DM/DO ratio in the cell [38]. Presumably, DM “editing” decreases the number of antigens, characterized by a low affinity for MHC-II, on the antigen-presenting cell (APC) surface, as was shown for DQ1 and DQ6 [39], as well as for DR3 alleles [31]. Analysis of the immunopeptidome of thymic APCs revealed that most antigens detected on MHC-II have a high affinity for MHC-II molecules [40,41]. Most likely, the effect of nonclassical MHC molecules is one of the important factors ensuring the highly competitive conditions for MHC-II ligand binding, which eventually results in the presentation of higher-affinity antigens.

It should be emphasized that the presentation of preferentially high-affinity antigens in the thymus may have its own shortcomings. It has been demonstrated that the MHC-II-presented antigen repertoire from tissues affected by autoimmune processes mostly consists of low-affinity peptides [42,43]. The abundance of self-antigens increases in peripheral tissues (e.g., myelin basic protein (MBP) in the central nervous system (CNS) [44,45,46] and insulin in the pancreas [47]). These protein self-antigens can be processed in the extracellular environment outside APCs. Next, the resulting antigenic fragments, characterized by a low affinity for MHC-II molecules, can be loaded directly on the APC surface or can enter early endosomes with low DM levels, thus escaping DM editing, and as a result, can be presented on the cell surface (Figure 3A,B). Certain MHC-II risk alleles bind CLIP with diminished affinity, which additionally promotes the binding of low-affinity antigenic peptides [48,49]. Rapid dissociation of CLIP enables the generation of empty complexes without DM involvement, which may potentially bind low-affinity antigenic peptides. Due to the spontaneous release of CLIP, low-affinity antigenic peptides are able to bind MHC-II in early endosomes without DM assistance or directly on the cell surface. Therefore, pMHC complexes on the APC surface at the periphery can activate TCRs that have not undergone negative selection due to the low abundance of this pMHC in the thymus. For MS patients, there was detected an immune response to full-length MS autoantigen proteolipid protein (PLP), naturally processed by APCs, with 2 immunodominant epitopes generation. Alternatively, the T cell response to several PLP synthetic fragments with high affinity to MS-associated MHC-II alleles was also observed after additional T cell stimulation by these peptides [50]. Since these fragments are normally hidden in a protein fold and are inaccessible for the endosomal proteases, the T cell activation seems to be the result of extracellular PLP processing and loading on MHC-II. Thus, the classical intracellular processing of antigen involves its capture by the APC with proteolytic processing and generation of fragments with optimal binding registers with the participation of HLA-DM in late endosomes, followed by subsequent exposure of pMHCs on the APC surface. These pMHCs take part in the negative selection process in the thymus, induce T cell clonal deletion and, after all, no autoreactive T cell response is observed at the periphery. Since pMHCs with “suboptimal” fragments were present in the thymus at low levels, autoreactive T cells might engage them at the periphery due to the failure of negative selection in the thymus.

Furthermore, a single antigen fragment can have several MHC-II binding registers. As T1D develops in non-obese diabetic (NOD) mice (the mouse model of T1D), autoreactive T cells recognize insulin (Ins. B:9–23) fragments presented on the product of diabetogenic allele I-A^g7^ [51,52]. The Ins. B:9–23 peptide has several binding registers in the context of I-A^g7^. The antigen with binding register 12–20 is characterized by low affinity and an increased dissociation rate in the presence of DM, while fragment 13–21 forms a more stable complex with I-A^g7^, which is almost insusceptible to DM editing. Immunization with antigen fragments 12–20 elicits an autoreactive T cell response, while immunization with antigen fragments 13–21 causes no response [53]. Presumably, the autoreactive TCR, specific to the antigen fragment 13–21, undergoes negative selection in the thymus, while the TCR, interacting with the antigen fragment 12–20, escapes the central tolerance mechanisms due to the potentially low abundance of the pMHC complex on the medullary APCs. Additionally, multiple autoreactive insulin-specific T cells were reported to recognize Ins. B:9–23 bound to I-A^g7^ in a low-affinity register 3 (Ins. B:14–22). The Ins. B:9–23 bound to I-A^g7^ activates diabetogenic CD4^+^ T cells and binds insulin-specific pancreatic T cells from NOD mice [54,55]. Thus, peptides may bind MHC-II molecules in low-affinity register in early endosomes or on the cell surface in the absence of DM due to the abundance of certain antigens in the autoimmunity-affected peripheral sites. Derived pMHC complexes engage autoreactive T cells escaping negative selection owing to the underrepresentation of unstable pMHCs in the thymus. Human DQ8 shares structural similarities with mouse I-A^g7^. The Ins. B:11–23 binds DQ8 with a low affinity and engages autoreactive peripheral blood CD4^+^ T cells of subjects with T1D [56].

Concluding, nonclassical MHC molecules help to present the most high-affinity antigens with the optimal binding register by MHC-II on thymic APCs. Nevertheless, when there is an excess of a certain self-antigen in peripheral tissues, its presentation on MHC-II molecules can occur without DM editing, allowing the presentation of low-affinity self-antigen fragments on the surface of APCs. The pMHC complexes loaded by low-affinity antigens are unstable and have a short lifespan; however, if their abundance on the surface of APCs is sufficiently high, they can potentially activate T cells, because the strength of the autoreactive T cell response is not directly correlated with antigen affinity to MHC-II [57].

## 3. The Peculiarities of the Interaction between TCRs and Self-pMHC Complexes: Links with Autoimmunity

Generally, CD4^+^ T cell TCRs recognize foreign antigens within MHC-II molecules [58,59,60] with CD4 receptors playing the key role in the induction of the T cell signaling cascade [61,62,63]. The analysis of trimolecular complexes between TCRs, MHC-II, and exogenous antigens has revealed some structural similarity in the TCR binding topology [64]. The TCR is arranged diagonally with respect to the antigen residing in the peptide-binding groove limited by α-helices. The variable regions of α and β TCR chains interact with the β and α chains of the MHC-II molecule, respectively [65]. The most structurally diverse CDR3 fragments of α and β TCR chains are located above the P5 residue of the bound peptide, while the CDR1 and CDR2 fragments interact with the α-helices of MHC-II molecules. The binding of TCR HA1.7 to the fragment of hemagglutinin HA protein within DR1 is an example of canonical interaction [66].

It was shown that, by contrast with TCRs binding bacterial or viral antigens on MHC-II, some autoreactive TCRs have an alternative binding topology. Thus, the TCR from an MS patient (Ob.1A12) specific for the fragment of myelin basic protein (MBP), which is presented on the HLA-DR2b risk allele molecule, interacts mainly with the N-terminus of the MBP fragment [16]. Furthermore, this TCR does not reside in the canonical diagonal position: its orientation angle is 110°, as opposed to 70° for HA1.7. The main interaction with the pMHC complex occurs due to the binding of the CDR3 region of TCR with the P2 residue of the MBP peptide fragment. Other TCRs, Ob.2F3 and Ob.3D1, obtained from the same donor with MS, also have an alternative binding topology, as examined using computational simulation [67]. Another TCR, 3A6, binds the MBP fragment in the complex with different DR15 allomorph DR2a, which is positively associated with the risk of developing MS. The 3A6 also has a suboptimal topology and low affinity for pMHC, similar to other autoreactive TCRs [68]. Its CDRs are shifted toward the N-terminus of the antigen, and CDR3 is also located above P2 of the MBP fragment. Nevertheless, similar to HA1.7, 3A6 binds diagonally with respect to pMHC. Therefore, Ob.1A12 and 3A6 have low affinity for pMHC, and their binding topology is alternate to anti-viral and anti-bacterial TCRs, which may potentially facilitate the escape from negative selection mechanisms in the thymus. Low-affinity TCRs were also reported in the pathogenesis of T1D. The repertoire of the autoreactive T cells, infiltrating pancreatic islets, exhibited low self-reactivity and promoted the development of T1D [69]. Presumably, the interaction between these T1D TCRs and self-antigen-carrying pMHCs is also characterized by alternate topology. The MBP fragment presented by DQ1 is recognized by the Hy.1B11 TCR with relatively high affinity [70]. These TCRs have canonical localization but are significantly shifted with respect to the DQ1α chain so that their interaction with the antigen is substantially limited. Only the CDR3 region of the Hy.1B11α chain contacts the MBP fragment. Presumably, the unusual topology of autoreactive TCRs binding to pMHC impedes activation by the CD4 molecule and, therefore, clonal deletion of T cells at the negative selection stage. This may explain the existence of T cells with similar TCRs (such as Hy.1B11) in the periphery [71]. The unusual topology of the trimolecular complex often implies the participation of a low-affinity TCR. However, autoimmune TCRs with a high affinity for pMHCs have also been reported. It is known that DQ molecules have a lower expression level than DR molecules. Therefore, appropriate autoreactive TCRs might require higher affinities for DQ pMHC complexes to overcome clonal deletion in the thymus [70]. Another possible explanation of high-affinity autoreactive TCR emergence at the periphery is that the autoantigenic fragment may be a weak binder in the pMHC complex. As previously discussed, pMHC carrying low-affinity peptides allows T cells to escape negative selection. Nevertheless, they can further bind autoreactive TCRs at the periphery. Interestingly, the weak interaction between the MBP fragment and the product of the risk allele DR4 is stabilized by the MS2-3C8 TCR involved in canonical high-affinity interaction with pMHC [72]. The structures of the trimolecular complexes of three autoreactive TCRs in model mice with experimental autoimmune encephalomyelitis (EAE), which bind the MBP fragment in complex with I-A^u^, have revealed canonical binding. However, the interaction between the MBP antigen and the I-A^u^ molecule is characterized by low affinity due to partial occupation of the peptide-binding cleft [73,74,75]. The affinities of self-peptides for MHC-II molecules of risk alleles and autoreactive TCRs for pMHC complexes of risk alleles are summarized in Table 1.

## 4. The Effect of MHC-II-Presented Self-Antigen Modifications on Autoreactive TCR Recognition

Antigens processed for further presentation by MHC-II molecules can undergo various posttranslational modifications (PTMs), including glycosylation, iodination, citrullination, etc. (Figure 4). PTMs can either occur spontaneously or be induced by various enzymes. Autoreactive T cell responses to modified self-antigens have been observed for many ADs. Supposedly, the modified antigens (neoantigens) are either totally not present in the thymus or their quantity is extremely low. These modifications presumably take place in peripheral tissues and are absent when the antigen is presented by mTEC cells in the thymus, which has been demonstrated for glycosylated type II collagen for the development of RA [76]. Although modified antigens are available in the thymus due to the presence of migratory APCs, their quantity is probably insufficient for negative T cell selection in the thymus. Thus, thyroglobulin, which is the autoantigen in patients with Hashimoto’s thyroiditis, is present in the thymus in its nonmodified form, whereas the level of its iodinated form, which seems to initiate the autoreactive T cell response, is extremely low in the thymus and is insufficient for central tolerance to be established [77].

Citrullination of antigens is described for a number of ADs, especially for RA; the positive charge of arginine is lost due to the conversion to citrulline, which ultimately alters the epitope affinity to MHC-II and TCR [78]. The products of risk MHC-II alleles for RA carry a consensus amino acid sequence in the antigen-binding groove, forming a positively charged P4 pocket–shared epitope (SE), which binds the polar amino acid residues of peptides at the P4 position (e.g., citrulline) and induces the activation of autoreactive T cells [79,80]. In addition to hosting a polar-neutral citrulline residue, SE can itself serve as the main contact zone for autoreactive TCRs and represent citrullinated fragments of the fibrinogen autoantigen that directly interact with autoreactive TCRs via the P2-citrulline residue [81]. Autoreactive T cells specific for citrullinated tenascin-C antigenic peptides, presented by DR4 molecules, were revealed in RA patients [82]. Additionally, glycosylation was reported to be responsible for the pathogenesis of RA. Analysis of the crystal structure of the trimolecular complex of an autoreactive TCR and type II collagen (Col2) neoantigen presented on DR4 has shown that lysine residues subjected to galactosylation are the key sites for TCR recognition [83]. Furthermore, the autoreactive T cell response is dependent on the galactosylation of lysine in the Col2_259–273_ fragment at position 264. Modified autoantigenic peptides also participate in T1D development. Several citrullinated and transglutaminated GAD65 epitopes are recognized by autoreactive CD4^+^ T cells in T1D preferentially to their unmodified form [84]. Autoreactive T cells specific for citrullinated glucokinase epitopes are linked to T1D pathogenesis [85]. Another example of the modified antigens impacting the pathogenesis of T1D is the generation of a disulfide bond. T cells recognize insulin fragments presented on DR4 molecules in T1D upon the assembly of a disulfide bridge between neighboring cysteine residues [86]. The citrullinated fragments of MBP and myelin oligodendrocyte glycoprotein (MOG) cause EAE due to the activation of autoreactive T cells [87,88,89]. In EAE, acetylation of the MBP fragment elicits an encephalitogenic T cell response [73,74,75].

Hybrid insulin peptides (HIPs) are identified as neoantigens involved in the pathogenesis of T1D [90]. HIPs are formed via a peptide bond formation between insulin fragments and other secretory granule peptides. Mass spectrometry analysis of pancreatic islets revealed the presence of HIPs in humans and mice [91]. The risk MHC-II molecules are reported to present various HIPs. The HIPs formed by proinsulin C-peptide and chromogranin A or islet amyloid polypeptide 2 (IAPP2) were present on I-A^g7^ molecules in the NOD mouse pancreas and recognized by pathogenic T cells [92]. Autoreactive T cells are able to recognize HIPs in the context of DQ2 and/or DQ8 and produce proinflammatory cytokines during the development of T1D [93,94]. The fusion of proinsulin C peptide and neuropeptide Y presented by DQ8+ and DR4+ B cells elicits an autoreactive T cell response in T1D [95]. DR4 presents different HIPs and is recognized by effector memory T cells [96]. The structures of trimolecular complexes of hybrid antigens formed by C-peptide and IAPP2, which are presented on DQ8, have revealed that the TCRα chain interacts exclusively with the C-peptide, while the TCR β chain interacts with IAPP2 only [97].

Multiple antigens with PTMs existing at the periphery cannot be inefficiently presented on thymic APCs, which results in the abrogation of autoreactive T cell clonal deletion. Concluding, PTM-modified antigens can be responsible for the induction of autoreactive T cells and for developing autoimmunity.

## 5. The Effect of the Proinflammatory Environment on MHC-II Antigen Presentation and Autoreactive T Cell Engagement

The generation and expansion of pathological T cells may be attributed to the increase in MHC-II expression levels due to the proinflammatory environment induced by viral or bacterial infections. The impacts of viral and bacterial infections on antigen presentation are the proposed triggers of pathological CD4^+^ T cell expansion in ADs such as MS and T1D [98,99,100,101,102]. Earlier, a proinflammatory environment was implicated in TCR-independent bystander activation in different ADs [103,104]. Proinflammatory cytokine release and bacterial antigen presence trigger the elevated synthesis of MHC-II molecules to maximize the presentation of foreign antigens for an efficient CD4^+^ T cell response. High IFN-γ levels increase the number of MHC-II molecules on professional and nonprofessional APCs, thereby altering the composition of the MHC peptidome in the inflammatory area [105]. Additionally, APCs, activated in pathogenic conditions, express an increased number of costimulatory molecules [106,107]. Therefore, rare self-antigen pMHC, which cannot induce an autoreactive T cell response under normal conditions, might engage autoreactive TCRs in a proinflammatory environment (Figure 5). Concluding, cytokines induced by the inflammatory environment may potentially activate antigen-specific autoreactive T cells.

A correlation between the inflammatory milieu and elevated MHC-II levels has been observed for several ADs. It was shown that MHC-II transcripts were upregulated by proinflammatory cytokine expression in β-cells of T1D patients [108,109]. IFN-γ induces the expression of I-A^g7^ molecules on beta cells of NOD mice, leading to the emergence of CD4^+^ autoreactive T cells and the promotion of T1D [110]. In addition, IFN-γ promotes the expression of MHC-II on islet endothelial cells in NOD mice [111]. The transcription factors responsible for the expression of MHC-II are elevated in MS lesions of human brain tissue [112]. Aberrant MHC-II expression was also found on oligodendrocytes in EAE mice and MS patients [113]. The expression of MHC-II in mouse joints led to the development of severe erosive inflammatory polyarthritis [114]. Additionally, the inflammatory environment promotes the expression of MHC-II molecules on hepatocytes during autoimmune hepatitis and pMHC transfer via trogocytosis to interacting CD4^+^ T cells, which further amplifies the autoimmune response [115].

The onset of autoimmunity often coincides with exposure to viral and bacterial infections. The T1D progression is probably linked with Coxsackievirus B, rotavirus, and other viruses [116,117,118]. MS development correlates with several infections, such as Epstein–Barr virus (EBV) and herpesvirus-6 [119,120,121]. The proinflammatory cytokines, potentially induced by infections, may reshape the repertoire of antigens presented on MHC-II in inflammation-affected sites. As previously discussed, some autoreactive T cells are characterized by altered binding topology to the cognate pMHCs, which results in inefficient clonal deletion of autoreactive T cells in the thymus. Therefore, autoreactive T cells mildly engaging with pMHCs under homeostatic conditions might develop an autoimmune response under inflammatory conditions due to the general expansion of pMHCs in the inflamed area.

## 6. Molecular Mimicry between Self- and Foreign MHC-II Antigens May Lead to T Cell-Mediated Autoimmunity

Molecular mimicry refers to a structural similarity between self and exogenous antigens and potentially leads to autoimmunity [122,123,124]. The structural resemblance of antigens might result in the occasional misactivation of T cells (Figure 6). Autoreactive T cells from MS patients specific for MBP can cross-react with EBV nuclear antigen 1 (EBVNA1) [125,126]. Apparently, the virus-specific TCR was able to bind structurally similar self-antigen fragments presented on MHC-II and initiate an autoreactive T cell response. The probability of MS development is significantly increased when EBV infection is combined with the MHC-II risk allele HLA-DRB1*15:01. It was shown that CD4^+^ T cells specific for DR15-presented EBV antigens could cross-react with MBP [127].

Narcolepsy is a rare autoimmune chronic neurological disorder positively associated with the HLA-DQB1*06:02 allele and characterized by targeting hypocretin neurons [128]. Cross-reactive T cells for hypocretin autoantigen (HCRT_NH2_) and flu HA (H1N1 2009 strain) antigens were shown to influence narcolepsy development [129]. T cell cross-reactivity between viral epitope neuraminidase (NA_175–189_) and self-epitope protein-O-mannosyltransferase 1 (POMT1_675–689_) is also involved in narcolepsy pathogenesis [130].

Anti-neutrophil cytoplasmic antibody (ANCA)-associated vasculitis (AAV) is an autoimmune disease targeting myeloperoxidase (MPO). Autoreactive T cells specific for the MPO_409–428_ cross-react with the 6PGD_391–410_ epitopes from *Staphylococcus aureus* [131]. Exogenous/self MHC-II antigen mimicry was also noted between streptococcal M protein and human cardiac myosin in the pathogenesis of rheumatic heart disease [132]. A bacterial L-asparaginase antigen fragment (L-ASNase67-81), mimicking a type II collagen epitope, may be presented on DR4 and elicits a CD4^+^ T cell response in blood samples of RA patients [133].

The gut microbiome potentially influences the development of various autoimmune pathologies [134,135]. It was shown that carrying certain risk or protective MHC-II alleles correlates with the composition of the gut microbiome in patients with AD [136,137,138]. Supposedly, distinct foreign antigenic peptides structurally similar to self-epitopes might cross-react with pathological T cells. Peptides from *Lactobacillus reuteri* mimic MOG antigenic fragments and elicit autoreactive T cell responses in an EAE model [139]. The DR53 (HLA-DRB4*01:03) molecule, presenting epitopes from *Roseburia intestinalis,* may bind beta-2 glycoprotein I (b2GPI) antigenic peptide and activate autoreactive T cells in antiphospholipid syndrome [140]. The antigenic peptide hprt4-18 from gut bacteria *Parabacteroides distasonis*, which is structurally similar to the fragment insulin B:9–23, activates human and NOD mouse insulin-specific T cells ex vivo [141,142].

Summarizing, bacterial and viral infections or gut microbiota composition seem to be the possible triggers of AD development through the mechanism of molecular mimicry, involving the recognition of structurally similar exogenous and endogenous antigenic fragments presented by MHC-II by CD4^+^ T cells.

## 7. The Role of Protective MHC-II Alleles in the Development of Autoimmune Diseases

Certain MHC alleles provide so-called “protection” toward the development of immune-related diseases due to peculiarities of antigen presentation, which affect the engagement of T cells [143,144]. “Protective” allele means that the risk of AD initiation in individuals carrying this allele is statistically lower than that for healthy donors, representing the average population [145,146,147,148]. Interestingly, the protein products of MHC-II protective alleles often differ from the products of risk alleles only by several amino acid residues localized in the TCR contact sites or near the key positions of the peptide-binding groove [149]. Supposedly, the protective effect can be ensured due to the deletion of autoreactive T cells in the thymus and/or induction of Treg cells (Figure 7A).

The expression of the I-E protective allele in NOD mice normally having the H-2^g7^ haplotype seems to prevent the development of T1D [150]. The protective effect of the I-E allele is reasoned by the deletion of autoreactive T cells [151]. Autoreactive TCR (4.1 TCR), expressed in transgenic NOD mice, undergoes negative selection in mice with the H-2^g7/b^, H-2^g7/k^, H-2^g7/q^, and H-2^g7/nb1^ haplotypes due to interaction with the products of protective MHC-II alleles [152]. Nevertheless, some studies cast doubt on the theory of deletion of autoreactive T cell clones as the only explanation for the protectivity of distinct MHC-II alleles in the development of ADs [153,154]. ADs are often characterized by a broadened repertoire of autoreactive T cells; therefore, protection against the development of autoimmune pathology cannot be solely attributed to the negative selection of a limited number of clones.

Tregs play a crucial role in maintaining peripheral tolerance and preventing ADs and tissue damage by controlling T cell expansion [155,156]. Supposedly, Tregs attenuate the immune response by engaging pMHCs encoded by protective alleles. However, the exact molecular mechanisms, involved in the interaction between Treg TCRs and the protective allele pMHC are still enigmatic. DR15 risk and DR1 protective molecules in Goodpasture syndrome present the α3 chain of type IV collagen (α3_135–145_) antigen with different binding registers [21]. Autoreactive T cells bound this self-antigen pMHC in DR15-positive patients with Goodpasture syndrome, and immunization with this antigen in DR15 humanized transgenic mice led to the development of the model disease. On the other hand, DR1 presenting α3_135–145_ interacts with Tregs in healthy donors. The DR1 humanized transgenic mice were similarly characterized by Treg differentiation and resistance to disease development. Presumably, changes in the register of antigen binding to the MHC-II molecules alter the antigen’s amino acid binding pattern, engaging TCR, which results in the predetermination of the phenotype of bound T cells.

Tregs may also play a role in the protection toward T1D development [157]. The protective (DQ6) T1D MHC-II molecules prevent the binding of islet-specific antigens to predisposing (DQ8) MHC-II molecules by engaging epitopes in different binding registers [158]. The proposed “epitope stealing” mechanism mediated by DQ6 molecules prevents the development of an autoreactive T cell response in T1D, elicited by autoantigen presentation on DQ8 molecules. It is possible that the antigenic peptides, presented by DQ6, promote the selection of Tregs in the thymus, which subsequently attenuates T1D development at peripheral sites. Healthy donors with the DR15/DQ6 haplotype had a higher level of T1D self-antigen-specific Tregs compared to those carrying neutral or risk MHC-II alleles [27]. Protective MHC-II alleles are also involved in the shaping of the gut microbial community and differentiation of Tregs. NOD mice, expressing the protective MHC-II Eα gene, do not develop T1D due to the change in gut microbiome composition and increased Treg proportions in the cecum [137].

Some studies revealed that autoreactive T cells can bind a number of antidiabetogenic MHC-II molecules, but differentiation into Tregs occurs due to engaging pMHCs of protective alleles exclusively [159,160]. The interaction with several pMHC complexes can be explained by TCR promiscuity and the unusual binding topology of some autoreactive TCRs, which was discussed previously. Interestingly, the trimolecular complex formed between Tregs and the proinsulin complex, presented by DR4 molecules, is characterized by 180° reversed polarity docking of TCR compared to canonical binding [161]. Thus, the TCRα and TCRβ chains interact with the α chain and β chain of the MHC-II molecule, respectively, while opposite interactions take place upon canonical binding. There is a limited number of reported crystal structures of trimolecular complexes of Tregs TCRs, making it difficult to accurately determine whether this binding topology is typical for most Tregs. However, canonical binding was revealed by analyzing the TCR structure of neonatal Tregs binding Padi4 peptide in the complex with I-A^b^ [162]. The crystal structures of trimolecular complexes of TCR and self-antigens presented by MHC-II protective alleles, which could possibly elucidate the molecular mechanism of protectivity, have not yet been reported.

The protective effect provided by MHC-II can also be related to the low rate of self-antigen loading to MHC-II molecules (Figure 7B). Antigen binding to MHC-II is a time-limited and competitive process controlled by nonclassical HLA molecules, as discussed above. It has been demonstrated that the self-antigen fragment MBP_151-164_ may be presented by the DR1 MHC-II allele, which is protective for patients with MS but is characterized by a slow loading rate in comparison with the exogenous peptide of influenza virus hemagglutinin protein HA_306–318_ [22]. This kinetic discrimination is structurally reasoned by decelerated docking of MBP residues in the P4 pocket. Taking into consideration the additional DM editing and the presence of other exogenous antigens competing with MBP peptide for DR1 in late endosomes, this fragment possibly cannot reach the surface of APCs. Therefore, the self-antigen MBP peptide is not presented or underrepresented by the protective MHC-II allele and does not elicit the possible autoreactive T cell response. However, the hypothesis of kinetic discrimination of self-antigens binding to molecules of protective MHC alleles needs additional clarification in terms of extended studies of MHC-II alleles relevant to the variety of ADs.

## 8. Conclusions

In this review, we systematized the data about MHC-II self-antigen presentation as an important pathway involved in the induction of the autoreactive CD4^+^ T cells. MHC-II allele polymorphisms are responsible for the diversity of the T cell repertoire. Normally, autoantigen-specific T cells are eliminated during negative selection in the thymus. However, pathologic autoreactive T cell clones can be expanded upon the development of ADs. Autoreactive T cells seem to appear due to the breakdown in the mechanisms of central tolerance in the thymus and may be activated as a result of interaction with pMHC carrying self-antigen on APCs. Normally, self-antigens, characterized by high affinity to MHC-II, are efficiently presented on APCs in the thymus and cause strong binding with the TCR of CD4^+^ T cells leading to the elimination of autoreactive T cells by negative selection. Self-antigens with low affinity for MHC-II molecules are limitedly presented in the thymus because of preliminary antigen repertoire editing by nonclassical MHC-II molecules. During inflammation, the abundance of self-antigens could be increased in the periphery (for example, in the inflammation locus), and low-affinity self-antigens can probably be loaded on MHC-II molecules without HLA-DM catalysis. In addition, many autoreactive TCRs have an unusual topology of binding to self-antigens presented on risk allele molecules, which can also potentially explain why autoreactive T cells escape negative selection in the thymus. The PTMs of autoantigens in peripheral tissue also may lead to recognition by T cells which have not undergone negative selection in the thymus. Finally, viral or bacterial infections might play a role in the development of ADs via the generation of a special proinflammatory environment or molecular mimicry between self- and pathogen antigens.

Here we have also discussed the presentation of self-antigens on protective MHC-II allele products and its significance for the subsequent induction of CD4^+^ T cell response. Summarizing, its protective role may be realized by the following scenarios: (i) deletion of autoreactive T cell clones, specific to protective pMHC in the thymus, during negative selection; (ii) induction of Treg development; and (iii) low rate loading of self-antigen on the protective MHC-II molecules and its kinetic discrimination in the concurrence with infection exogenous antigens.

Future determination of the etiology of ADs and steps toward targeted therapeutic approaches require the identification of a broader repertoire of self-antigens capable of binding presentation to the products of risk and protective MHC-II alleles along with the identification of pathogenic TCRs connected with the recognition of auto-pMHC. On the other hand, the basis of the thymic negative selection is directly linked with the MHC-II-mediated presentation of self-antigens; therefore, balancing between central and peripheral tolerance versus autoreactivity may be caused by thermodynamic and kinetic peculiarities of the autoantigen loading on the MHC-II molecules. Elucidation of the molecular mechanisms underlying the assembly of the autoreactive trimolecular pMHC-II-TCR complexes and its further signaling may result in the elaboration of novel therapeutic modalities in terms of the induction of the immunological tolerance [46,163,164,165], topological blockade of the self-antigens [166,167,168], and specific depletion of the autoreactive T cell clones [169,170].

## Figures and Tables

**Figure 1 cells-12-00314-f001:**
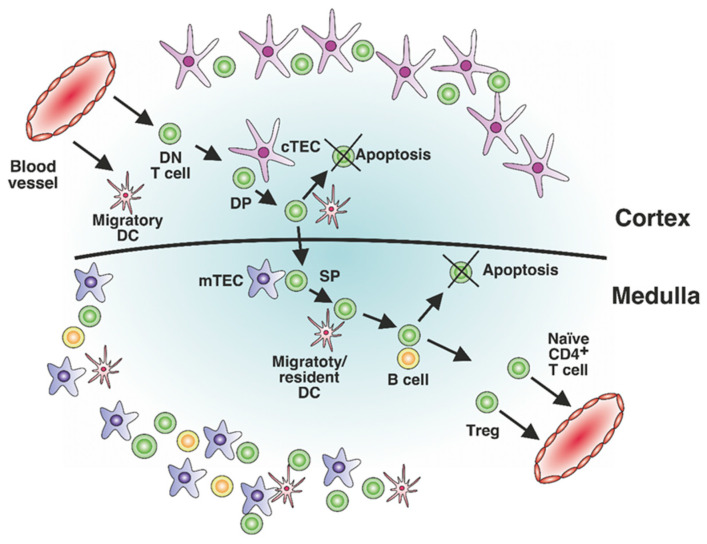
Stages of T cell selection during the establishment of central tolerance. T cell precursor double-negative (DN) T cells enter the cortex of the thymus through a blood vessel to undergo positive selection. Due to the interaction between antigen-presenting cells (APCs) present on cortical thymic epithelial cells (cTECs) and migratory dendritic cells (DCs), double-positive (DP) T cells are differentiated into CD4^+^ or CD8^+^ single-positive (SP) T cells and migrate to the medulla of the thymus to undergo negative selection. As they interact with APCs (medulla thymic epithelial cells (mTECs), migratory or resident DCs, and B cells), CD4^+^ T cells become CD4^+^ naïve T cells or T regulatory cells (Tregs) and leave the thymus. Negative selection is schematically shown only for SP CD4^+^ T cells.

**Figure 2 cells-12-00314-f002:**
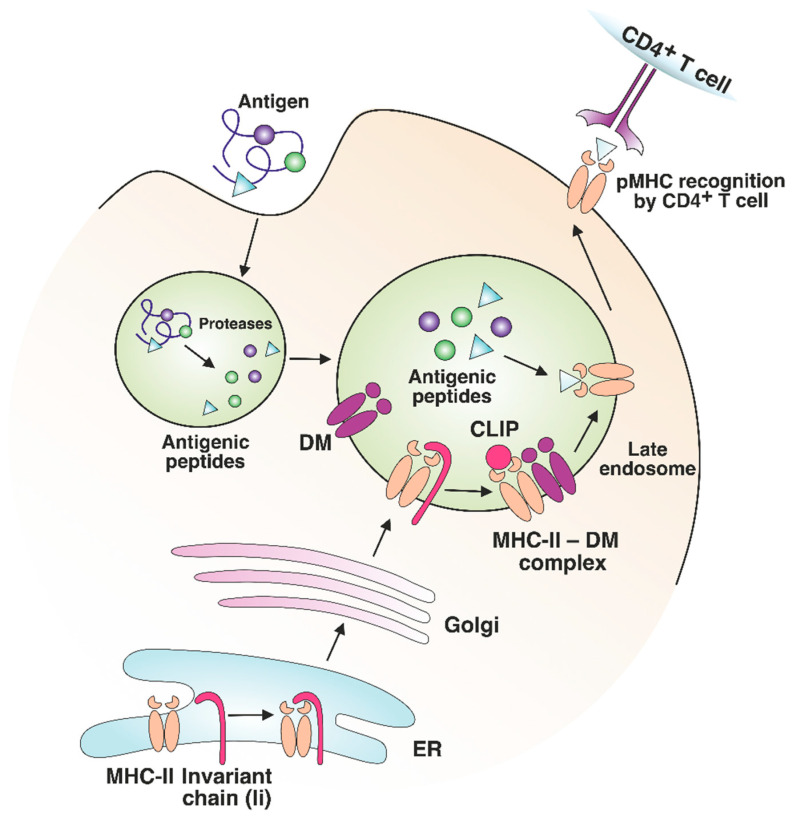
MHC-II maturation and antigenic peptide loading. MHC-II molecules are synthesized in the endoplasmic reticulum (ER) and loaded with an invariant chain (Ii). The MHC-II complex with Ii is transported through the Golgi to the late endosome. Endosomal proteases process antigens to short peptides and Ii to shorter class II-associated invariant chain peptide (CLIP). The binding of a nonclassical HLA-DM (DM) molecule to MHC-II promotes CLIP exchange to the antigenic peptide with an optimal binding register. The formed pMHC complex is transported to the surface of the APC for CD4^+^ T cell recognition.

**Figure 3 cells-12-00314-f003:**
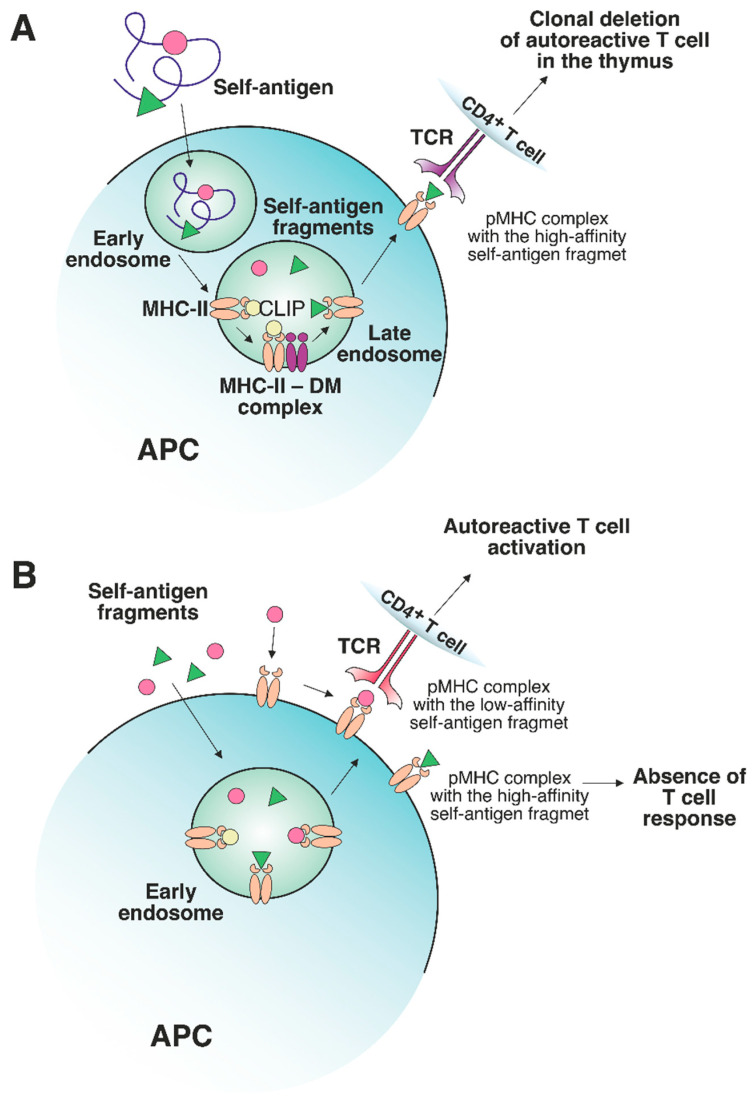
The role of a nonclassical DM molecule in the editing of the self-antigen repertoire presented by MHC-II. (**A**) In a thymic APC, the self-antigen is processed through a classical pathway under exposure to cellular proteolytic enzymes. Next, CLIP is exchanged to the high-affinity fragment in late endosomes under DM control. The stable pMHC complex is then transferred to the APC surface, where the T cells specific to this pMHC undergo clonal deletion. (**B**) If an excessive amount of certain self-antigens, processed outside APCs, is present in peripheral tissues, then these self-antigen fragments can be loaded on MHC-II molecules in early endosomes or on the surface of APCs without the recruitment of DM. The absence of DM editing allows the emergence of either stable or unstable pMHC complexes. The pMHCs, containing high-affinity fragments, do not elicit a T cell response due to the deletion of autoreactive T cells in the thymus. Unstable complexes with low-affinity fragments can bind autoreactive T cells since such T cells are not susceptible to clonal deletion.

**Figure 4 cells-12-00314-f004:**
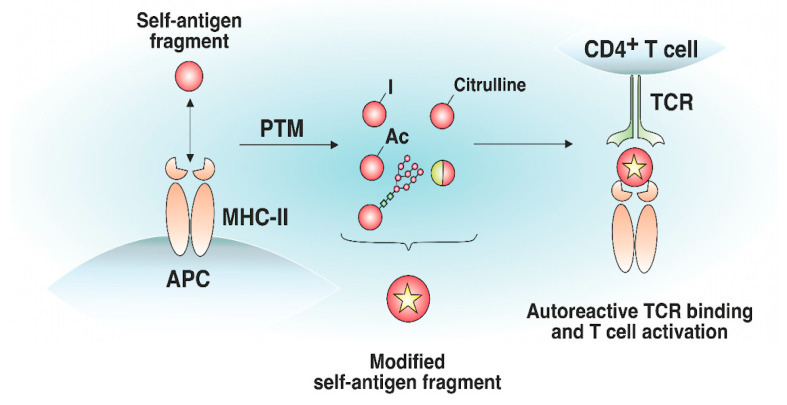
Effect of posttranslational modifications of self-antigens on trimolecular complex assembly. Certain self-antigen fragments do not bind to MHC-II molecules, or their binding affinity is low. Similar self-antigen with posttranslational modifications (iodination, acetylation, glycosylation, citrullination, generation of hybrid peptides, etc.) exhibiting increased affinity for MHC-II molecules, can appear in peripheral tissues. The resulting pMHC complexes can bind autoreactive T cells that have not undergone negative selection in the thymus.

**Figure 5 cells-12-00314-f005:**
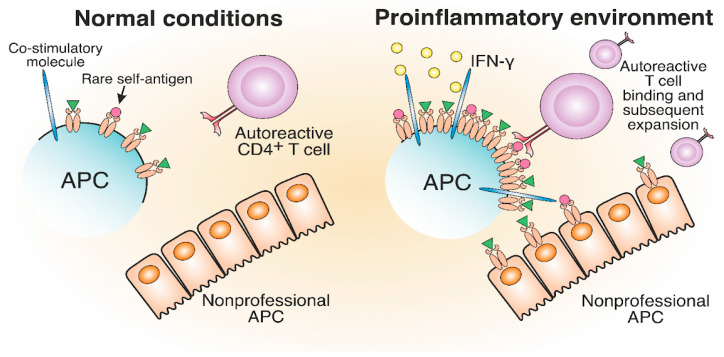
The development of autoreactive CD4^+^ T cells in a proinflammatory environment. The proinflammatory environment caused by viral or bacterial infection entails IFN-γ production, which results in increased MHC-II and costimulatory molecule expression compared to normal conditions. Therefore, rare self-antigens are present in a greater number, which can lead to the potential activation of autoreactive T cells.

**Figure 6 cells-12-00314-f006:**
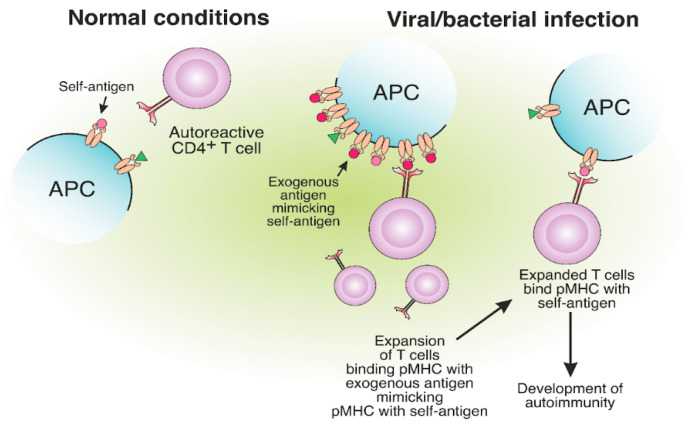
The expansion of autoreactive CD4^+^ T cells due to the molecular mimicry between foreign and self-antigens. The presentation of viral or bacterial antigenic peptides on MHC-II molecules, structurally similar to self-antigens, may result in the expansion of CD4^+^ T cells, specific to exogenous pMHC. Expanded CD4^+^ T cells, specific for foreign peptides, can bind the self-antigen pMHC, which results in the development of autoimmunity.

**Figure 7 cells-12-00314-f007:**
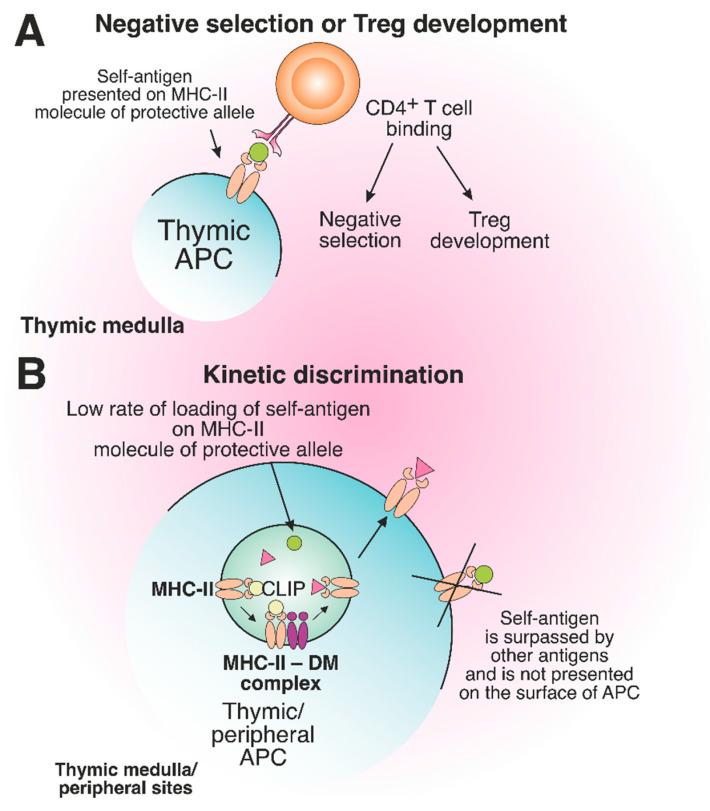
The role of protective MHC-II molecules in the suppression of autoreactive T cell development. (**A**) Recognition of the self-antigen presented on the product of the protective allele by CD4^+^ T cell results in negative selection or Treg development in the thymus. (**B**) The slow loading rate of self-antigen on the MHC-II molecule of the protective allele results in the failure of pMHC complex assembly on the surface of thymic or peripheral APCs.

**Table 1 cells-12-00314-t001:** Autoantigens and autoimmune TCRs in autoimmune diseases.

AD	Species	MHC-II Allele	Antigenic Peptide	Proposed Molecular Mechanisms of AD Susceptibility	Reference
Low-affinity autoantigenic peptides					
T1D	Mouse	I-A^g7^	Ins. B:12–20, Ins. B:13–21	Weak binding of CLIP by MHC-II molecule promotes loading of low-affinity peptides.	[48]
T1D	Mouse	I-A^g7^	Ins. B:12–20, Ins. B:14–22	MHC-II binding of antigenic peptide in a low-affinity register leads to recognition by autoreactive TCRs.	[52,53,54,55]
T1D	Human	HLA-DQA1*03:01, HLA-DQB1*03:02	Ins. B:11–23	MHC-II binding of antigenic peptide in a low-affinity register leads to recognition by autoreactive TCRs.	[56]
Low-affinity autoreactive TCRs					
MS	Human	HLA-DRB1*15:01	MBP:85–99	Autoreactive TCRs bind pMHC complexes with low affinity and alternate topology.	[16,67]
MS	Human	HLA-DRB5*01:01	MBP:84–102		[68]
High or intermediate affinity autoreactive TCRs					
MS	Human	HLA-DQA1*01:02, HLA-DQB1*05:02	MBP:85–99	The unusual topology of trimolecular complex impedes T cell activation by CD4 receptor.	[70]
MS	Human	HLA-DRB1*04:01	MBP:111–129	Autoreactive TCR stabilizes weak interaction between antigenic peptide and MHC-II.	[72]
EAE	Mouse	I-A^u^	MBP:1–11 (acetylated)	Autoreactive TCRs bind pMHC complexes with acetylated autoantigens in unusual binding register, where the part of peptide-binding groove is empty.	[73,74,75]

Abbreviations: AD, autoimmune disease; T1D, type 1 diabetes; Ins, insulin; CLIP, class II-associated invariant chain peptide; MS, multiple sclerosis; MBP, myelin basic protein; TCR, T cell receptor; EAE, experimental autoimmune encephalomyelitis.

## Data Availability

Not applicable.

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
