# Peer review of "MHC Class II Presentation in Autoimmunity"

_cells, 2023, doi:10.3390/cells12020314_

Round 1

Reviewer 1 Report

 Y. Zakharova et al. have summarized the information on the MHC-II presentation of self-antigens in some of the most relevant autoimmune-mediated diseases with special emphasis on the molecular mechanisims involved. In addition to describing the basic concepts on the molecules mediating activation of autoreactive T cells, this review also includes specific evidence of the relevance of MHC-II in the whole process.

I hope the following comments will help them to improve the manuscript.

1. Abbreviations must be specified the first time they appear in the text. Therefore, it should be indicated here on line 77 that “Ads” stands for autoimmune diseases. The abbreviation EAE in line 307 had already appeared in line 255, so it should be used directly without re-specifying its description.  Abbreviations should also be described in the figure caption. In Figure 2, the abbreviations CLIP and DM molecule must be specified. In this regard, the title of Table 1 should be corrected, and a footnote should be included in the table explaining the abbreviations included therein.

2. Lines 164-180: This is a very complex paragraph in its wording. It provides a lot of data that would be much clearer if it included a concluding sentence that summarises the main idea that this data is supposed to support.

3. Lines 173-174:  Please correct the format. There is a paragraph break where it should not be

4. Lines 244-259:  Please correct the format.

5. The format of Table 1 needs to be improved. The columns do not correspond to their heading and should try to summarise the text included.

6. Line 299: At this point in the manuscript, there is a transition from one disease to another (RA to TD1) without any kind of connector to anticipate the reader, so it is confusing to find information organised in this way.

7. Lines 357-360: The authors should specify whether the data they provide in the information extracted from references 11 to 114 are obtained in mouse models or in patients.

Author Response

The author collective expresses their appreciation and gratitude to the reviewers for their careful and detailed consideration of our article.

All reviewers’ comments were taken into account while working with the text.

In particular, comments 1-7 of reviewer 1 (Note: line numbering has changed due to insertions and formatting)

  1. All required decoding of abbreviations at the first mention has been included (line 81, line 255), including in the captions of figures 1 and 2 and table headings
  2. Lines 185-189: a concluding sentence, summarizing the main idea of the paragraph, is included in the text.
  3. Lines 173-174: format is corrected
  4. Lines 244-259: format is corrected
  5. The format of Table 1 is improved. We reformulated the headings of the columns and included text.
  6. Line 314: the sentence introducing T1D is included
  7. Lines 372-377: the information from references 111 to 114 - clarified the object of research (mouse models or patients).

Reviewer 2 Report

This is an interesting and well written review regarding MHC II presentation and autoimmunity. The figures and tables are well prepared and illustrative. 

Author Response

The author collective expresses their appreciation and gratitude to the reviewers for their careful and detailed consideration of our article.

We hope that the changes made in accordance with the recommendations of the reviewers have improved the presentation of our manuscript.

Reviewer 3 Report

This review systematically summarizes the research on MHC-II self-antigen presentation as an important pathway involved in the induction of self-reactive CD4+ T cells. This provides more comprehensive information and new ideas for us to understand and further study the role of MHC-II in autoimmunity. Manuscript is well written and the topic is very relevant and interesting. A small suggestion, it is best to describe the structure, function and biological activity of MHC-II in detail in the introduction part.

Author Response

The author collective expresses their appreciation and gratitude to the reviewers for their careful and detailed consideration of our article.

All reviewers’ comments were taken into account while working with the text.

We have taken into account the suggestion of reviewer  and have supplemented the introduction section with two sentences connected with MHC II presentation mechanism (line 56-59). More detailed information about MHC II presentation mechanism is given in section 2 (lines 94-105) and Figure 2.

We hope that the changes made in accordance with the recommendations of the reviewer have improved the presentation of our manuscript.